# Associations between hospital deaths (HSMR), readmission and length of stay (LOS): a longitudinal assessment of performance results and facility characteristics of teaching and large-sized hospitals in Canada between 2013–2014 and 2017–2018

Omid Fekri, Edgar Manukyan, Niek Klazinga

University Medical Centers, University of Amsterdam, Amsterdam, Noord-Holland, Netherlands

**Correspondence to**
Omid Fekri;
o.fekri@amsterdamumc.nl

## ABSTRACT

**Objectives** To examine the association between hospital deaths (hospital standardised mortality ratio, HSMR), readmission, length of stay (LOS) and eight hospital characteristics.

**Design** Longitudinal observational study.

**Setting** A total of 119 teaching and large-sized hospitals in Canada between fiscal years 2013–2014 and 2017–2018.

**Participants** Analysis focused on indicator results and characteristics of individual Canadian hospitals.

**Primary and secondary outcomes** Hospital deaths (HSMR); all patients readmitted to hospital; average LOS and a series of eight hospital characteristic summary measures: number of acute care hospital stays; number of acute care beds; number of emergency department visits; average acute care resource intensity weight; total acute care resource intensity weight; hospital occupancy rate; patients admitted through the emergency department (%); patient days in alternate level of care (%).

**Results** Comparing 2013–2014 to 2017–2018, hospital deaths (HSMR) largely declined, while readmissions increased; 69% of hospitals decreased their hospital deaths (HSMR), while 65% of hospitals increased their readmissions rates. A greater proportion of community-large hospitals (31%, n=14) improved on both hospital deaths (HSMR) and readmission compared to Teaching hospitals (13.9%, n=5). Hospital deaths (HSMR), readmission and LOS largely showed very weak and non-significant correlations. LOS was largely positively and statistically significantly correlated with the suite of eight hospital characteristics. Hospital deaths (HSMR) was largely negatively (not statistically significantly) correlated with the hospital characteristics. Readmission was largely not statistically significantly correlated and showed no clear pattern of correlation (direction) with hospital characteristics.

**Conclusions** Examining publicly reported hospital performance results can reveal meaningful insights into the association among outcome indicators and hospital

### Strengths and limitations of this study

► Assessed correlations across eight hospital characteristics and three hospital performance indicators.
► Assessed 5 years of performance data.
► Examined the majority of teaching and community-large hospitals in Canada.
► Inability to apply more complex statistical modelling techniques due to limitations on the use of aggregate hospital-level data in secondary analyses.
► Length of stay is an aggregate of all hospitalisations, and could not be restricted to condition-specific cases (of hospital death or readmission).

characteristics. Good or bad hospital performance in one care domain does not necessarily reflect similar performance in other care domains. Thus, caution is warranted in a narrow use of outcome indicators in the design and operationalisation of hospital performance measurement and governance models (namely pay-for-performance schemes). Analysis such as this can also inform quality improvement strategies and targeted efforts to address domains of care experiencing declining performance over time; further granular subdivision of the analyses, for example, by hospital peer-groups, can reveal notable differences in performance.

## INTRODUCTION

Over the last two decades, there has been substantial interest in hospital performance,[1] and with financing of hospitals increasingly tied to improving the quality of care delivered.[2] Along with improving the quality of care, a tandem goal of hospital reforms has been to improve efficiency[3] (ie, reducing waste, streamlining care pathways, increasing patient throughput, optimising the use of



technology, etc). Hospital deaths[4] and readmission to hospital[5] are among the most commonly used indicators to measure quality of hospital care, while average length of stay (LOS) is often used as a measure of efficiency.[6] The three measures together (hospital deaths, readmission and LOS) have been the subject of increased interest in recent years to assist with more reliable interpretations of hospital performance.[7]

However, the goals of achieving quality and efficiency can at times be opposing. For example, it seems warranted to investigate whether a hastened hospital stay (shorter LOS) would lead to an increased chance of readmission to hospital.[8] Similarly, do efforts to reduce hospital readmissions have the unintended consequence of increasing the likelihood of mortality after hospitalisation?[9] While hospital deaths and readmission are both desired to be reduced, it is not definite (and varying across diseases and clinical procedures) whether a patient's LOS should be lower or higher in order to minimise readmission or in-hospital mortality. However, what can be deduced is that the relationships between LOS, in-hospital mortality and readmission are intertwined and interdependent. Hence, governance of hospitals based on these publicly reported indicators should be based on acknowledgement and consideration of these interdependencies.

Yet, despite a sizeable research community investigating the interrelationship between these indicators, the evidence base on the patterns of these interdependencies remains inconclusive due to wide heterogeneity in methods and findings across studies (which speaks to the complexity of the topic). For example, a switch between the unit of analysis (from patient level to hospital level), on the same underlying admissions data, will yield inconsistent, and even inverse, results.[10] In recent years, researchers have also examined hospital characteristics, such as hospital volumes[11] or hospital teaching status[12] to better understand any associations between LOS, readmission and in-hospital mortality.

Much of the afore cited literature originates from the USA and Europe. With a scarcity of local examples, this study used a large, nationally representative dataset of hospital performance measures (produced by the Canadian Institute for Health Information (CIHI)) to expand interest and add evidence for the Canadian context. Specifically, we investigate the relationship between hospital deaths, readmission and LOS, and explore any associations with hospital characteristics. Our specific research questions are:

1. How have hospitals performed in both the hospital deaths (hospital standardised mortality ratio, HSMR) and readmission indicators over time?
2. What is the correlation between hospital deaths (HSMR), readmissions and LOS?
3. How do a series of eight hospital characteristics correlate with hospital deaths (HSMR), readmissions and LOS?

4. Do the results of the aforementioned research questions show differences between peer groups of teaching hospitals and community-large hospitals?

## METHODS
### Data
We used the all data export report file from CIHI's Your Health System In Depth online tool[13] to perform the analyses. The data file contains results per hospital for all indicators published on the online tool as well as contextual measures and additional variables to assist with analysis and interpretation. Five singleton fiscal year (1 April to 31 March) data points were available covering 2013–2014 to 2017–2018 for the indicators capturing hospital deaths (HSMR) and all patients readmitted to hospital (henceforth referred to 'readmission'), while LOS and eight hospital characteristics measures were only available for the most recent year (2017–2018).

### Definition of variables
The following indicators were used for the analysis: hospital deaths (HSMR), readmission (%) and LOS (days); and eight contextual measures of hospital facility characteristics: number of acute care hospital stays; number of acute care beds; number of emergency department visits; average acute care resource intensity weight (RIW); total acute care RIW; hospital occupancy rate; patients admitted through the emergency department; patient days in alternate level of care (%).

HSMR and other variations of summary hospital mortality measures are commonly used indicators to assess hospital performance. The hospital deaths (HSMR) indicator is a ratio of observed to expected in-hospital mortality, capturing the 72 leading causes of hospital death (representing ~80% of all in-hospital mortality). The Readmission indicator captures all urgent patient readmissions within 30 days. The average LOS indicator is a sum of all valid days spent in hospital, divided by the total number of inpatient cases. Detailed technical notes on these indicators,[14] and on hospital facility characteristics,[15] are made available by CIHI through its Indicator Library.

Both hospital deaths (HSMR) and readmission indicators are risk adjusted. Hospital deaths (HSMR) risk-adjustment variables are: age, sex, LOS, admission category, comorbidity (Charlson Index Score) and transfers. As the readmission indicator is an aggregate of four subcategories of readmission (medical, surgical, obstetric, paediatric), the readmission risk-adjustment variables are not constant across the four subcategories; this range of risk-adjustment variables are: age, sex, acute care hospitalisations in previous 6 months, admission category, comorbidity (Charlson Index Score) and casemix groupings. Detailed information on model specifications and coefficients used in calculations are available elsewhere.[16 17]

CIHI classifies the approximately 600 hospitals in Canada into four distinct peer-group types: teaching

hospitals; community-large hospitals; community-medium hospitals and community-small hospitals. This classification facilitates meaningful comparisons across hospitals of similar structural characteristics, patient volume and clinical complexity.[18] Since characteristics of hospitals are not included in risk-adjustment models, any comparison of two or more hospitals' individual performance should be done within their respective hospital peer-groups.

A hospital is designated as 'teaching' by provincial/territorial ministries of health, or was identified as such in the provincial/territorial ministry's submission to CIHI's Management Information System Database. Community-large hospitals meet two of the following three criteria: more than 8000 inpatient cases; more than 10 000 weighted cases; or more than 50 000 inpatient days.

In order to qualify for public reporting of results for the hospital deaths (HSMR) indicator, a hospital must meet a minimum of 2500 eligible hospital deaths (HSMR) cases for each of the most recent three consecutive years.[19] Consequently, no community-small hospitals met this criteria to have publicly reported hospital deaths (HSMR) results. Of the 93 community-medium hospitals only 11 hospitals met the minimum reporting requirements and had hospital deaths (HSMR) results reported. Since this represents only 8.5% of the entire peer-group, it was decided to also exclude community-medium hospitals, alongside community-small hospitals, in this analysis. Hospitals with only 1 year of data available, for both-readmission and hospital deaths (HSMR) indicators, for either 2013–2014 or 2017–2018 only, were excluded from performance trend analysis. Therefore, a total of 119 hospitals were included in the overall study, 53 Teaching hospitals and 66 community-large hospitals (representing 67.9% and 68.2% of all hospitals in their respective peer-group totals in the available online dataset). A subset of 81 hospitals were included in the performance trend analysis.

## Statistical analyses

Descriptive statistics for the analysis of LOS, hospital deaths (HSMR) and readmission indicators are presented by range of values, peer-group means and 95% CIs and coefficient of variation (CoV) (see table 1). Trend over time is calculated as the percent-change difference between first and last year of data (2013–14 and 2017–18). A paired t-test was used to determine whether absolute changes in rates between 2013–2014 and 2017–2018 were significant.

To compare indicator rates per hospital across 2013–2014 to 2017–2018, three possible outcomes are inferred: a decrease in rate (2013–2014>2017–2018); an increase in rate (2013–2014<2017–2018); and no change in rate (2013–2014=2017–2018). Multiplying these three outcomes by the two indicators of interest (hospital deaths (HSMR) and readmission), in tandem, yields a total of nine trend outcomes (see table 2).

Graphical representation of the aforementioned tests are shown via scatterplots depicting: (1) percent change

**Table 1** Descriptive statistics for combined analysis of Hospital Deaths (HSMR), Readmission and LOS

| | Teaching hospitals | | | | Community-large hospitals | | | |
|---|---|---|---|---|---|---|---|---|
| No of hospitals, n | 36 | | | | 45 | | | |
| Range of values for 2017–2018 data year | Range of values | Teaching peer-group mean* (95% CI) | Coefficient of variation, % (95% CI) | Median (IQR Q1–Q3) | Range of values | Community-large Peer-group mean* (95% CI) | Coefficient of variation, % (95% CI) | Median (IQR Q1–Q3) |
| LOS (days) | 4.6 to 9.2 | 7.1 (6.7 to 7.4) | 16 (13 to 21) | 6.9 (6.4–7.8) | 4.5 to 13.7 | 6.5 (6.1 to 6.9) | 24 (20 to 29) | 6.2 (5.7–7.1) |
| Hospital deaths (HSMR) | 66 to 118 | 91.8 (87.8 to 95.7) | 14 (11 to 18) | 92 (82–100) | 65 to 144 | 87.5 (83.9 to 91) | 16 (13 to 19) | 86 (77–96.5) |
| Readmission (%) | 7.4 to 10.6 | 9.4 (9.2 to 9.6) | 8 (7 to 11) | 9.5 (9–9.9) | 7.4 to 10.7 | 8.9 (8.7 to 9.1) | 8 (7 to 10) | 8.8 (8.5–9.63) |
| Per cent-change difference 2013–2014 vs 2017–2018 (%) | Range of % change | Mean teaching peer-group % change* (95% CI) | | | Range of % change | Mean community-large peer-group % change* (95% CI) | | |
| Hospital deaths (HSMR) | –21 to 22 | –4.1 (–7.5 to –0.8) | | | –33 to 21 | –6.0 (–9.1 to –2.8) | | |
| Readmission (%) | –12 to 12 | 2.1 (0.7 to 3.6) | | | –14 to 17 | 1.6 (–0.3 to 3.4) | | |

*Calculated by summing values of all hospitals within peer-group and dividing by number of hospitals
HSMR, hospital standardised mortality ratio; LOS, length of stay.

**Table 2** Performance trend outcomes on Hospital Deaths (HSMR) and Readmission (2013-2014 to 2017-2018)

| Trend outcome | Hospital deaths (HSMR) | Readmission | Teaching hospitals (total n=36) No, (%) | Community-large hospitals (total n=45) No, (%) | Total of all hospitals, no, (%) |
|---|---|---|---|---|---|
| Decrease in both HSMR and readmission | ↓ | ↓ | 5 (13.9) | 14 (31.1) | 19 (23.5) |
| Decrease in HSMR, increase in Readmission | ↓ | ↑ | 20 (55.6) | 14 (31.1) | 34 (42.0) |
| Decrease in HSMR, no change in readmission | ↓ | = | 1 (2.8) | 2 (4.4) | 3 (3.7) |
| Increase in HSMR, decrease in readmission | ↑ | ↓ | 2 (5.6) | 1 (2.2) | 3 (3.7) |
| Increase in both HSMR and readmission | ↑ | ↑ | 7 (19.4) | 8 (17.8) | 15 (18.5) |
| Increase in HSMR, no change in readmission | ↑ | = | 1 (2.8) | 0 | 1 (1.2) |
| No change in HSMR, decrease in readmission | = | ↓ | 0 | 1 (2.2) | 1 (1.2) |
| No change in HSMR, increase in readmission | = | ↑ | 0 | 4 (8.9) | 4 (4.9) |
| No change in both HSMR and readmission | = | = | 0 | 1 (2.2) | 1 (1.2) |

* ↑ =signifies increasing rate; ↓ =signifies decreasing rate; =signifies no change

over time for hospital deaths (HSMR) and readmission (delineated by peer-group) (see figure 1) and (2) 2017–2018 data year results on hospital deaths (HSMR) and readmission, with LOS depicted as the size of the bubble plot (see figures 2 and 3).

A Spearman's rank correlation test examines the association between LOS, hospital deaths (HSMR) and readmission on 2017–2018 data year values (with breakdowns for teaching and community-large hospital peer-groups). Strengths of correlations, the absolute value of $R_s$ (positive and negative) are defined as: 0.00–0.19 very weak; 0.20–.39 weak; 0.40–0.59 moderate; 0.60–0.79 strong; 0.80–1.0 very strong.[20]

Lastly, a Spearman's rank correlation test was also used to assess the correlation between eight hospital facility characteristics against LOS, hospital deaths (HSMR) and readmission values for 2017–2018. All analyses were performed on R V.3.5.0 (R Foundation for Statistical Computing, Vienna, Austria).

### Patient and public involvement
Patients or public were not involved in the design of this longitudinal, observational study. However, all data used are available in the public domain.

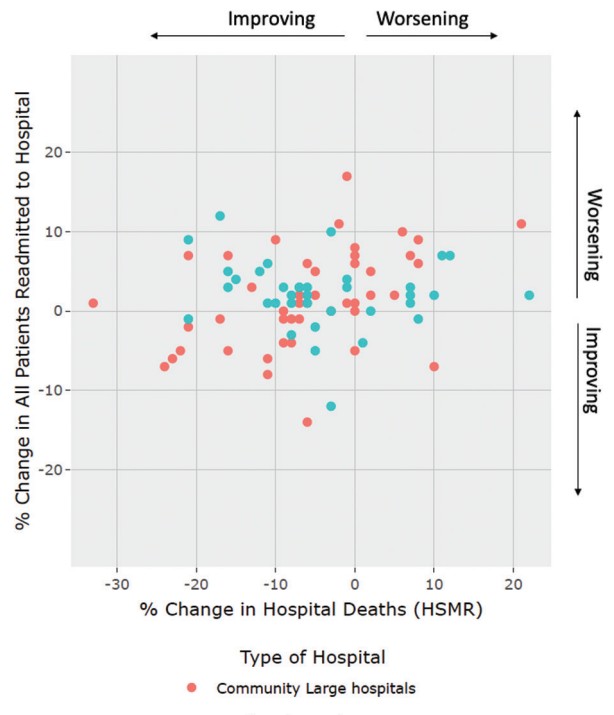

**Figure 1** Scatterplot of percent-change between 2013–2014 and 2017–2018 for Readmission and Hospital Deaths (HSMR) (by hospital peer-group). HSMR, hospital standardised mortality ratio.

### RESULTS
#### Combined performance of hospital mortality (HSMR) and readmission over time
In comparing 2013–2014 and 2017–2018 indicator rates, hospital deaths (HSMR) largely declined, while readmissions increased (see table 1). A paired t-test showed statistically significant changes in trend over time for both indicators: hospital deaths (HSMR) improved by a mean of −5.1 (95% CI −7.33 to −2.9, p<0.001), and readmission rates increased by a mean of 0.15% (95% CI 0.04% to 0.26%, p=0.006). While not statistically significant, the community-large hospital peer-group showed a greater mean improvement in hospital deaths (HSMR) by −6.0% (95% CI −9.1% to −2.8%), while teaching hospitals improved by −4.1% (95% CI −7.5% to −0.8). Both hospital peer groups experienced a mean increase in readmission rates, with community-large hospitals at 1.6% (95% CI −0.3% to 3.4%) and teaching hospitals at 2.1% (95% CI 0.7% to 3.6%). When examining the 2017–2018 data year, community-large hospitals had a statistically significant lower rate of readmissions at 8.9 (95% CI 8.7 to 9.1) compared with teaching hospitals at 9.4 (95% CI 9.2 to 9.6). Table 2 provides a lens on how individual hospitals performed in both indicators. Nine possible outcomes

 Fekri O, *et al. BMJ Open* 2021;**11**:e041648. doi:10.1136/bmjopen-2020-041648

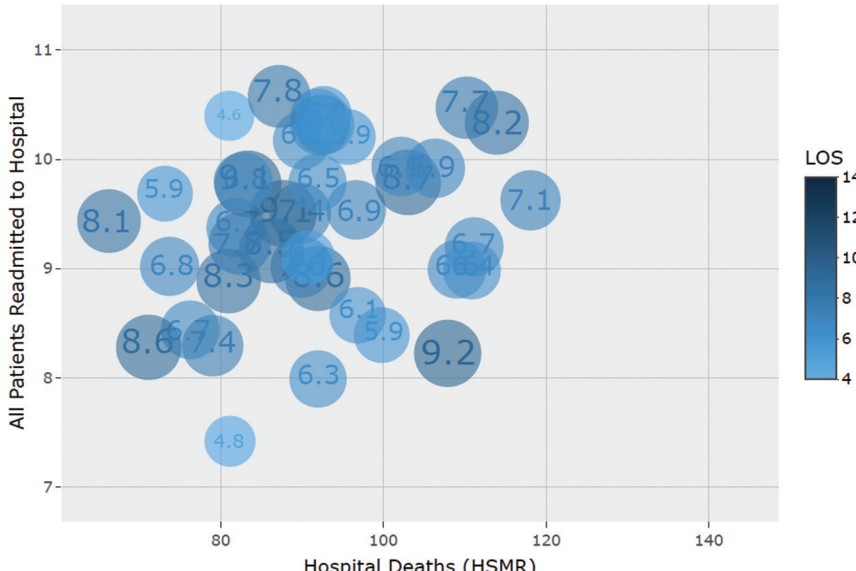

**Figure 2** Scatterplot of teaching hospital values for Hospital Deaths (HSMR), Readmission and LOS (2017–2018). HSMR, hospital standardised mortality ratio; LOS, length of stay.

of performance are shown. Overall, 56 (69%) out of the total 81 hospitals assessed decreased their hospital deaths (HSMR), while only 23 (28%) hospitals decreasing their readmissions rates.

Figure 1 illustrates the combined percent change of hospital deaths (HSMR) and readmissions rates (comparing 2013–2014 and 2017–2018 individual hospital rates) delineated by hospital peer group. While coefficient of variation values are largely similar between the two peer-groups for the two outcome indicators, nearly three times as many community-large hospitals (n=14) showed greater improvement in the bottom left quadrant

of figure 1 (decrease in both hospital deaths (HSMR) and readmission), than teaching hospitals (n=5). These clear trends of overall decreasing hospital deaths and rising readmissions have been confirmed in our previous analysis.[21]

### Hospital deaths (HSMR), readmissions and LOS (2017–2018)

In examining hospital deaths (HSMR), readmission and LOS for potential associations, only very weak to weak non-statistically significant results were observed (see table 3). The community-large hospital peer-group showed greater variation in LOS values (CoV=24%, 95%

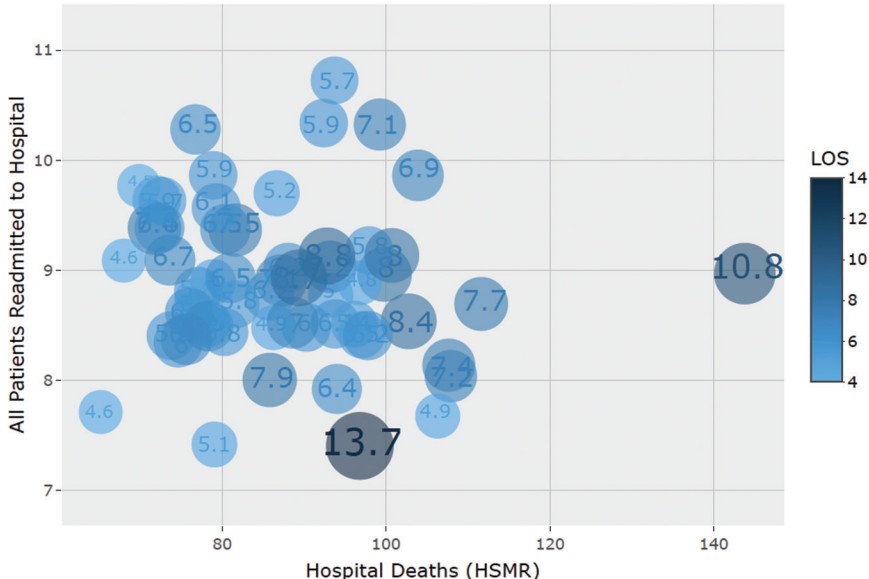

**Figure 3** Scatterplot of community-large hospital values for Hospital Deaths (HSMR), Readmission and LOS (2017–2018). HSMR, hospital standardised mortality ratio; LOS: length of stay.

**Table 3** Correlations between hospital deaths (HSMR), readmission and LOS (breakdowns by teaching and community-large hospitals) (2017–2018)

| | LOS | | | Hospital deaths (HSMR) | |
|---|---|---|---|---|---|
| **Readmission** | Teaching: | −0.04 (−0.41 to 0.33) | | Teaching: | 0.22 (−0.09 to 0.54) |
| | Community-large: | 0.04 (−0.23 to 0.31) | | Community-large: | −0.13 (−0.42 to 0.15) |

*Direction of correlation is shown as blue (positive) and red (negative) and intensity of cell-colouring reflects strength of correlation. Correlation strength classification: 0.00–0.19 very weak; 0.20–0.39 weak; 0.40–0.59 moderate; 0.60–0.79 strong; 0.80–1.0 very strong.
HSMR, hospital standardised mortality; LOS, length of stay.

CI 20% to 29%) compared with the teaching hospital peer-group (CoV=16%, 95% CI 13 to 21). While not statistically significant, the community-large hospital peer group had a shorter mean LOS of 6.5 days (95% CI 6.1 to 6.9) compared with the teaching hospital peer group of 7.1 days (95% CI 6.7 to 7.4) (see table 1). Figures 2 and 3 illustrate LOS, hospital deaths (HSMR) and readmission values for the 2017–2018 data year (with LOS delineated in size and shading of bubble plot) for teaching and community-large hospitals respectively.

### Correlation between hospital characteristics, LOS, hospital deaths (HSMR) and readmission

Table 4 shows the correlation between hospital characteristics and LOS, hospital deaths (HSMR) and readmissions. LOS was largely positively correlated (and statistically significant) with the series of eight hospital characteristics. Hospital deaths (HSMR) was largely weak to very weakly negatively correlated. Readmissions were mixed with positive and negative weak to very weak correlations. Correlations between hospital deaths (HSMR) and readmissions with the eight hospital characteristics were largely not statistically significant (aside from patient days in alternate level of care, patients admitted through the emergency department and average acute care RIW).

The number of acute care hospital stays was only statistically significantly correlated with LOS (negatively weakly) in community-large hospitals (r=−0.36, 95% CI −0.59 to −0.13, p<0.01). Teaching hospitals had a moderate positive and statistically significant correlation in the number of acute care beds and LOS (r=0.5, 95% CI 0.23 to 0.76, p<0.01). The number of emergency department visits and LOS were negatively moderately correlated in community-large hospitals (r=−0.44, 95% CI −0.7 to −0.17, p<0.01). The average acute care RIW was positively strongly correlated with LOS (r=0.68, 95% CI 0.56 to 0.8, p<0.01) when assessing both hospital peer groups. With respect to hospital deaths (HSMR), the average acute care RIW was positively moderately correlated in community-large hospitals (r=0.53, 95% CI 0.32 to 0.74, p<0.01). Total acute care RIW was only moderately positively correlated with LOS for teaching hospitals (r=0.43, 95% CI 0.06 to 0.7, p<0.01). Hospital occupancy rate was only statistically significantly correlated with LOS for teaching hospitals (r=0.37, 95% CI 0.07 to 0.67, p<0.05). With respect to hospital deaths (HSMR), a hospital's occupancy rate is very weak to weakly negatively correlated (and not statistically significant). Patients admitted through the

emergency department had a positive weak to moderate correlation with LOS (teaching hospitals r=0.47, 95% CI 0.18 to 0.75, p<0.01; community-large hospitals r=0.39, 95% 0.16 to 0.61, p<0.01) and a positive weak correlation with readmissions (teaching hospitals r=0.29, 95% CI 0 to 0.58, p<0.05; community-large hospitals r=0.27, 95% CI 0.03 to 0.52, p<0.05). The percentage of patient days in alternate level of care (a measurement of days patients spend in inpatient acute care, when unneeded, while waiting for discharge to home care or other supports are ready) had a positive weak correlation with LOS in Teaching hospitals (r=0.36, 95% CI 0.06 to 0.66, p<0.05), and a weak negative correlation with readmissions for all hospitals combined (r=−0.29, 95% CI −0.5 to −0.09, p<0.01).

Online supplemental data files include descriptive statistics (mean/per cent change values, CIs, range of values and number of hospitals) by indicator, facility characteristics, provincial/territorial jurisdiction, and hospital type/size, and correlation matrix scatterplots.

### DISCUSSION

In recent years, there has been growing interest in the association between hospital deaths, readmission and LOS.[7] It is logical to investigate the strength and directionality of correlation between these three components of hospital performance, and with hospital characteristics. There is wide heterogeneity in the available evidence in this research area. Aside from the natural differences across studies that narrow their scope in terms of disease or procedure-specific indicators, limited clinical settings within hospitals, and small denominator groups, even a change in the unit of analysis on the same underlying data, from patient-level data to hospital-level data, can yield disparate results.[10]

This secondary analysis of hospital performance data aimed to provide a high level overview of the association between hospital deaths, readmission and LOS across a majority of teaching and community-large hospitals in Canada between 2013–2014 and 2017–2018. The classification and assignment of hospital peer groups allows for more meaningful and valid comparisons of performance of hospitals across similar structural characteristics, patient volumes and clinical services offered. Therefore, any comparison of individual hospital performance should be restricted to within a respective peer-group.

**Table 4** Correlations between hospital characteristics on LOS, Hospital Deaths (HSMR) and readmission (2017–2018)

| Hospital characteristic | Unit | LOS Correlation coefficient (95% CI) | | Hospital deaths (HSMR) Correlation coefficient (95% CI) | | Readmission Correlation coefficient (95% CI) | |
|---|---|---|---|---|---|---|---|
| # of acute care hospital stays | # of days | All: | -0.04 (-0.24 to 0.16) | All: | -0.14 (-0.34 to 0.05) | All: | 0.07 (-0.12 to 0.26) |
| | | Teaching: | 0.26 (-0.02 to 0.54) | Teaching: | -0.30 (-0.61 to 0.01) | Teaching: | 0.07 (-0.23 to 0.37) |
| | | Community large: | -0.36*(-0.59 to -0.13) | Community large: | -0.20 (-0.45 to 0.05) | Community large: | -0.11 (-0.36 to 0.15) |
| # of acute care beds | # of beds | All: | 0.24* (0.05 to 0.42) | All: | -0.01 (-0.20 to 0.19) | All: | 0.03 (-0.16 to 0.22) |
| | | Teaching: | 0.50* (0.23 to 0.76) | Teaching: | -0.24 (-0.54 to 0.07) | Teaching: | -0.03 (-0.35 to 0.29) |
| | | Community large: | -0.02 (-0.24 to 0.20) | Community large: | 0.01 (-0.25 to 0.26) | Community large: | -0.17 (-0.41 to 0.07) |
| # of emergency department visits | # of visits | All: | -0.13 (-0.37 to 0.10) | All: | 0.03 (-0.21 to 0.27) | All: | 0.04 (-0.18 to 0.27) |
| | | Teaching: | 0.17 (-0.20 to 0.55) | Teaching: | -0.14 (-0.53 to 0.26) | Teaching: | 0.18 (-0.16 to 0.52) |
| | | Community large: | -0.44*(-0.70 to -0.17) | Community large: | 0.13 (-0.20 to 0.46) | Community large: | -0.20 (-0.49 to 0.09) |
| Average RIW | Average RIW | All: | 0.68* (0.56 to 0.80) | All: | 0.39* (0.20 to 0.57) | All: | 0.15 (-0.04 to 0.35) |
| | | Teaching: | 0.55* (0.31 to 0.80) | Teaching: | 0.00 (-0.31 to 0.31) | Teaching: | 0.12 (-0.20 to 0.45) |
| | | Community large: | 0.76* (0.62 to 0.89) | Community large: | 0.53* (0.32 to 0.74) | Community large: | -0.20 (-0.44 to 0.05) |
| Total RIW | Total RIW | All: | 0.13 (-0.06 to 0.33) | All: | -0.02 (-.22 to 0.17) | All: | 0.13 (-0.06 to 0.32) |
| | | Teaching: | 0.43* (0.16 to 0.70) | Teaching: | -0.25 (-0.55 to 0.06) | Teaching: | 0.11 (-0.20 to 0.41) |
| | | Community arge: | -0.16 (-0.40 to 0.08) | Community arge: | -0.06 (-0.32 to 0.19) | Community arge: | -0.13 (-0.39 to 0.12) |
| Hospital occupancy rate | % of occupancy | All: | 0.09 (-0.12 to 0.29) | All: | -0.14 (-0.37 to 0.08) | All: | 0.01 (-0.20 to 0.23) |
| | | Teaching: | 0.37 (0.07 to 0.67) | Teaching: | -0.28 (-0.61 to 0.06) | Teaching: | 0.00 (-0.34 to 0.34) |
| | | Community large: | -0.12 (-0.39 to 0.14) | Community large: | -0.10 (-0.41 to 0.21) | Community large: | 0.01 (-0.27 to 0.29) |
| Patients admitted through the emergency department | % of patients | All: | 0.30* (0.13 to 0.48) | All: | -0.11 (-0.31 to 0.08) | All: | 0.12 (-0.08 to 0.31) |
| | | Teaching: | 0.47* (0.18 to 0.75) | Teaching: | -0.04 (-0.41 to 0.32) | Teaching: | 0.29 (0.00 to 0.58) |
| | | Community large: | 0.39* (0.16 to 0.61) | Community large: | -0.10 (-0.36 to 0.16) | Community large: | 0.27 (0.03 to 0.52) |
| Patient days in alternate level of care | % | All: | 0.23 (0.03 to 0.43) | All: | -0.01 (-0.24 to 0.22) | All: | -0.29*(-0.50 to -0.09) |
| | | Teaching: | 0.36 (0.06 to 0.66) | Teaching: | 0.02 (-0.37 to 0.42) | Teaching: | -0.28 (-0.62 to 0.05) |
| | | Community arge: | 0.24 (-0.04 to 0.52) | Community large: | 0.07 (-0.27 to 0.40) | Community large: | -0.13 (-0.43 to 0.17) |

*P<0.01; ¹P<0.05; direction of correlation is shown as blue (positive) and red (negative) and intensity of cell-colouring reflects strength of correlation. Correlation strength classification: 0.00–0.19 very weak; 0.20–0.39 weak; 0.40–0.59 moderate; 0.60–0.79 strong; 0.80–1.0 very strong.

HSMR, hospital standardised mortality ratio; LOS, length of stay; RIW, resource intensity weight.

Delineating the results of this study's analyses by teaching and community-large hospitals allows for a more granular interpretation of hospital performance at peer-group level.

Of the three outcome indicators, only with the readmissions indicator was there a statistically significant result of community-large hospital peer-group showing a lower peer-group average than that of the teaching peer-group.

Detailed data on eight hospital characteristics were also available in the dataset published by the data steward. As this study was exploratory in nature, we additionally included these hospital characteristics in the correlation analyses to explore any meaningful relationships with the aforementioned three main indicators, and delineated by hospital peer-group type.

Our earlier research[21] established that, over time, Canadian hospitals have largely improved on in-hospital mortality; readmission rates have been trending upward; and that good or bad performance in one domain of care does not automatically reflect the same performance in other domains. What this present study aimed to add is whether a hospital's improvement or weakening performance over time, in either hospital deaths (HSMR) or readmission, had a positive or negative association on the other; our results showed that 42% of hospitals, the largest proportion across the possible outcomes, in fact decreased hospital deaths (HSMR) while increasing readmission rates. Furthermore, we added LOS to the research question as a proxy of hospital efficiency. Eight hospital characteristics showed trends in strength and directionality of correlation with hospital deaths (HSMR), readmission and LOS. As this study was exploratory in nature, in both using aggregate hospital-level data and hospital characteristics in the analyses, we did not have an explicit hypothesis on the degree of association between hospital characteristics and the three outcome indicators. We note (and continued to include in the analyses) an outlier hospital (see figure 3) with a high hospital deaths (HSMR) indicator value, a long LOS, and average readmission rate.

### Strengths and limitations of this study

The main strengths of this study are the quality and extent of data used; all teaching and community-large hospitals across Canada that had publicly available reported performance results were included in the analysis. The 'all readmission' indicator captures, as the title suggests, all readmission to hospital within 30 days; the hospital deaths (HSMR) indicator captures ~80% of all in-hospital mortality; and the LOS indicator quantifies the mean duration across all hospitalisations. Eight diverse hospital characteristics also provided summary measures that capture numerous aspects of a hospital's performance context. While results for LOS and the eight hospital characteristics were only available for the most-recent year (2017–2018), for hospital deaths (HSMR) and readmission indicators, five fiscal year data points were available to measure trend over time differences.

There are limitations in this study with respect to its generalisability beyond Canada; differences in risk-adjustment methodologies, indicator definitions and calculation methods, and hospital type/size definitions, pose challenges to make apples-to-apples comparisons across countries. However, the categorical outcomes of performance simultaneously comparing hospital deaths and readmission, along with the correlation tests of these indicators and hospital characteristics, is available and worthwhile to other settings. Community-medium and community-small hospitals in Canada treat fewer patients, and offer less-complex clinical services. This large group of hospitals (comprising more than half within the country) are omitted from this study due to an absence of publicly reported indicator values for hospital deaths. Furthermore, as a result of mergers between disparate hospitals, historic indicator values (ie, 2013–2014 data year) are omitted from the reporting platform. Thus, this inhibits a longitudinal comparison (ie, performance trend over time). However, current indicator values and hospital characteristics data are available and was included in analyses that only required 2017–2018 data year (namely, correlation analyses on hospital characteristics).

An important limitation of this study, inherent to the constraints of using aggregate-level hospital data, is the inability to perform more complex analyses. Previous, more granular analyses by researchers have been able to employ more sophisticated statistical techniques, including modelling, controlling for confounding factors, calculation of composite indicators, application of more refined case inclusion/exclusion criteria and stratification of analyses across different disease groups. Another such example of a limitation exists with the LOS measure reflecting the average of all hospitalisations, and the inability to select just those applicable to hospital deaths (HSMR) or readmission patients, respectively. Acknowledging these limitations of performing secondary analyses on aggregate, publicly available hospital performance data, we nonetheless pursued our four research questions, with the data available at hand, to determine what, if any, level of association exists at the hospital indicator level.

The two main outcome indicators themselves, hospital deaths (HSMR) and readmission, also have methodological limitations due to the inability of including non-hospital death data. The hospital deaths (HSMR) indicator, unlike the summary hospital-level mortality indicator, can only account for deaths that occur in hospitals. Similarly, the readmission indicator cannot exclude patients from the denominator that have passed away in the community following hospital discharge. While the indicators of hospital deaths (HSMR) and readmission are risk adjusted (as described in the Methods section), not all risk-factors can be adjusted for (due to reasons such as viability).[22] For example, detailed data on patient sociodemographics or access to primary care services is unavailable for risk-adjustment modelling. Lastly, as correlation

does not equal causation, the correlation-based results of this study should be interpreted with caution.

## Reflections on the study's findings

Public reporting of performance results poses challenges to hospital administrators and the broader public. Public reporting has become a staple in health systems and hospital performance management. But the practice of public reporting is not without concerns.[23] Tunnel vision and myopia by hospital governance and performance managers can run the risk of suboptimisation; the unintended consequences of shifting concentration disproportionately towards areas prioritised for immediate measurement at the expense of other areas of care and broader/long-term organisational goals.[24]

Pay for performance schemes are commonplace in hospital governance. A governance model that assesses hospitals through isolated performance measures, runs the risk of unintended consequences in other factors of care and performance not under immediate scrutiny.[8] The results and methods of this study support the notion that quantification of hospital performance should not be done via isolated or single measures at a time, but rather in a more broad and informed mechanism of considering complementary aspects of hospital performance (such as those in the CIHI hospital performance framework: access to services, clinical effectiveness, safety, coordination of care, patient-centredness and hospital efficiency).[25] Furthermore, a poorly conceptualised pay-for-performance scheme may be mal-aligned to take into consideration the correlation (and potential causality) of intensifying efforts to reduce, for example, LOS or hospital mortality, on the increase of readmission rates.

Moreover, government officials charged with hospital governance must take into account inequality across hospital facilities and hospital corporations. Beginning in the 1990s, but increasing rapidly in recent years, there has been a trend of mergers between multiple hospitals and between hospitals and rehabilitation institutes into a singular hospital corporation.[26] These larger hospital corporations in turn have near-exclusive coordination of care between acute care patients served in hospitals and subsequently their transfer to rehabilitation services. Rural and more-remote hospitals (especially those without paired rehabilitation services) could face higher LOS and occupancy rates, greater number of days and percentage of patients in alternate level of care, and greater resource utilisation. If analysis of these amalgamated hospitals and rehabilitation services proves they perform better than hospitals without direct rehabilitation services, this consideration should also be included in the contextual interpretation (and perhaps risk adjustment) of hospital performance and governance. Similarly, readmission to hospital may also be a proxy of the strength and availability of primary healthcare services in the community. Thus, the necessity to consider hospital performance in the broader context of an integrated health service delivery system, a tenet of the accountable care organisation movement.[27]

Government bodies and professional associations charged with supporting quality improvement initiatives can use the methods and findings of this type of analysis to identify best practices and top-performing hospitals so as to learn from their effective practices. Similarly, hospitals in an unfavourable quadrant (long LOS, and high hospital mortality and readmissions) should receive tailored programmes to support their improvement in quality and efficiency of care.

The general public, too, requires consideration when publicly reporting performance results. Efforts in describing indicators in plain language and providing a framework for contextualisation can increase the public's assimilation of performance results (especially demographic groups with fewer skills or resources).[28] CIHI's applies these practices in their online YHS tool, providing their health system performance[29] and hospital performance frameworks[25] as a basis for the curation of performance results, and describing both performance indicators and hospital characteristics in plain language.

The results of this study do not provide a definitive outcome to the debate on the complementarity between LOS, hospital deaths, readmission and hospital characteristics. The underlying pathways and differences between hospitals in functions, and scope of services provided, makes the hospital a complex unit of analyses. The corpus of past studies illustrates the wide heterogeneity of research methods and degree of association outcomes. The embedding of this type of analysis into hospital governance formulation can only better-inform those charged with policy-making and administrators of hospitals. Subdividing the research methods of this study, into disease and/or procedure-specific analysis, can help facilitate addressing quality improvement concerns on specific clinical areas; but caution is stressed so as to not unintentionally cause clinicians and hospital administrators to experience tunnel vision.

## CONCLUSIONS

This study shows that secondary analyses of publicly reported hospital performance results can reveal meaningful insights into the association among outcome indicators and hospital characteristics. Good or bad hospital performance in one care domain does not necessarily reflect similar performance in other care domains. Thus, caution is warranted in a narrow use of outcome indicators in the design and operationalisation of hospital performance measurement and governance models (namely pay-for-performance schemes). Analysis such as this can also inform quality-improvement strategies and targeted efforts to address domains of care experiencing declining performance over time; further granular subdivision of the analyses, for example by hospital peer-groups, can reveal notable differences in performance.

**Correction notice** This article has been corrected since it first published. The provenance and peer review statement has been included.

**Contributors** OF initially conceived of the study, reviewed the literature, performed data analysis, interpreted results and drafted the manuscript. EM assisted in the design of the study, performed and validated data analysis, interpreted results and reviewed the manuscript. NK assisted with the design of the study, interpreted results and assisted in the drafting of the manuscript.

**Funding** The authors have not declared a specific grant for this research from any funding agency in the public, commercial or not-for-profit sectors.

**Competing interests** None declared.

**Patient consent for publication** Not required.

**Provenance and peer review** Not commissioned; externally peer reviewed.

**Data availability statement** Data are available in a public, open access repository. All hospital performance and characteristics data used in this study is publicly available via CIHI's Your Health System online tool (http://yourhealthsystem.cihi.ca/).

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
