## [Reviewer comments · BMJ Open]

ARTICLE DETAILS

TITLE (PROVISIONAL)	Associations between Hospital Deaths (HSMR), Readmission and Length of Stay (LOS): a longitudinal assessment of performance results and facility characteristics of teaching and large-sized hospitals in Canada between 2013-14 to 2017-18
AUTHORS	Fekri, Omid; Manukyan, Edgar; Klazinga, Niek

VERSION 1 – REVIEW

REVIEWER	SANDRO SCARPELINI UNIVERSITY OF SÃO PAULO BRAZIL
REVIEW RETURNED	17-Jul-2020

GENERAL COMMENTS	Very good and important work. In particular, for the suggestion of a "new paradigm" for the future of the high complexity hospital's organization. As the author says that this new model has been developed since 1990, we can not forget that the study was done in Canada. In low in middle income countries, we are so far for starting this type o new organization, which transform this paper in a very interesting reference for these countries.
---

REVIEWER	Mariana Lobo Center for Health Technology and Services Research (CINTESIS) and Department of Community Medicine, Information and Health Decision Sciences (MEDCIDS), Faculty of Medicine, University of Porto, Porto, Portugal
REVIEW RETURNED	17-Aug-2020

GENERAL COMMENTS	The paper addresses a relevant aspect of hospital performance assessment regarding intertwined relationship of different performance indicators and the importance of acknowledging it by policy-makers. The presentation of results is very appealing and well thought. Major comments: While the paper presents a strong motivation, well-referenced, to use the indicators HSMR, readmission, and LOS, assessing their association as performance measures, the focus on other hospital characteristics, in my opinion, lacks a proper motivation. Particularly, why they have compared the two types of hospitals (community vs. Teaching hospitals) is not explored in the introduction nor is discussed in the section "Reflection's on the study findings". After reading this section it seems more relevant to compare hospital merges vs. hospital not merged and rural/remote area vs. urban hospitals.
--

	In the trend analysis of HSMR and readmissions, why did you use just the difference between the first and last period available and not the longitudinal analysis using several measures per hospital over the 5 years? Would that give you different results? One of the strengths of the study is the quality and comprehensiveness of the data used, which includes several indicators and covers most of the Canadian hospitals. However, in my opinion, the data has also important limitations that have not been addressed in the discussion. Specifically,  • The indicator All Patients Readmitted to Hospital does not control for the hospital case mix, and therefore, this could explain for example why teaching hospitals have a larger increase regarding readmissions; or could affect the association between indicators. • I am not familiar with the readmissions indicator, but I believe that if a patient dies during the 30-day period after discharge, the hospitalization will be counted in the denominator of this indicator. However, since this is a competing risk it cannot be counted in the numerator since the patient can no longer be readmitted. This may attenuate the increase of readmissions. • In-hospital deaths may not be the best performance indicator. It is certainly important for the period the patient is in the hospital, but 30-day mortality is considered a more suited indicator since the death during this period is likely to be associated with the care provided at the hospital. There seems to be an outlier community hospital in Figure 3, long stays, high death rate, and average readmission rate. Do you know why? Could you please elaborate or make a remark on this case in the discussion. Does it influence the correlation estimate? Minor comments: Page 8, Line 15: “A paired-t test showed statistically significant changes in trend over time for both indicators: hospital deaths (HSMR) improved by a mean of -5.1 (95% CI, -7.33 – -2.9, t=-4.58, df=80, p<.001). And readmission worsened by a mean of 0.15% (95% CI, 0.04 – 0.26, t=2.81, df=80, p=.006).” Does this refer to all hospitals? If yes, I think it would be better in line 2 of this page. Table 2 would read better if both the first and second columns were sorted by "green arrow", "red arrow" and "equal sign". As is, only the first column is sorted. Even though it is stated in the methods, I recommend adding the reference period of the analysis (2017-18) in the subtitle of the results in Page 10, line 36, and on Page 12, line 1. For comparisons between Figures 2 and 3 the shade of the markers (based on the LOS) should have the same scale. It seems that an average LOS of 7.7 in figure 2 is darker than in Figure 3. Please provide the abbreviation for RIW in table 4, page 12. Resource Intensity Weight? I prefer to see the strengths and limitations at the bottom of the discussion, but it may be just a personal preference.
--	--

REVIEWER	Pascal, Léa Hospices Civils de Lyon, France
REVIEW RETURNED	21-Aug-2020
GENERAL COMMENTS	Overall comment

	Using aggregated hospital-level data from 119 teaching and large-sized Canadian hospitals, Fekri O. and colleagues evaluated the evolution of hospital death and readmission between 2013-14 and 2017-18. They also studied the correlations of eight hospital characteristics with hospital death, readmission, and length of stay (LOS) in 2017-18. They concluded that care must be taken in the use of performance indicators to establish public health policies, as a hospital may have good performance with one indicator but poor performance with another. The subject is interesting, and a strength of this study is the use of publicly reported hospital performance indicators at the national level. However, important details concerning the methodology used for producing the hospital performance indicators are lacking while they are necessary to the interpretation of results. I also have concerns about the robustness of the methodology and the rigor in the interpretation of some results. Lastly, I would like the authors clarify what this study adds to the literature and in particular to a previous study published on the same data concerning one of their four research questions [Fekri O, Manukyan E, Klazinga N. Appropriateness, effectiveness, and safety of care delivered in Canadian hospitals: a longitudinal assessment on the utility of publicly reported performance trend data between 2012-2013 and 2016-2017. BMJ Open 2020; O:e035447. Doi:10.1136/bmjopen-2019-035447]. Please find below the detailed comments, hoping that my suggestions will help the authors to improve their paper. Major comments Methodological details behind the indicators and implications for the interpretation Outcomes used in this study are not precisely defined in the manuscript, and the reader is invited Page 6 Line 31 to consult the technical notes of the Canadian Institute for Health Information (CIHI) (references 14 and 15) from which the performance indicators originated. It is important for the reader to have a straightforward information about the inclusion and exclusion criteria and the models and covariates used, and to be able to take these elements into account when interpreting the results and making comparisons. 1. I recommend the authors to precise in the Methods section that hospital death and readmission are risk-adjusted, and to give the covariates used in the calculation, i.e.:  • for hospital deaths (HSMR): diagnosis groups, age, sex, LOS group, admission category (urgent and elective), comorbidity (Charlson Index Score) group and transfers (with reference to Canadian Institute for Health Information. Model Specifications — Clinical Indicators, May 2020. Ottawa, ON: CIHI; 2020), • for readmission: age, sex, acute care hospitalization in previous 6 months, admission category, comorbidity, and a selection of case-mix groups (CMG) according to patient groups considered (obstetric, pediatric, surgical, medical), with reference to Canadian Institute for Health Information. Hospital Standardized Mortality Ratio - Technical Notes, September 2019. Ottawa, ON: CIHI; 2019. 2. Given that HSMR is standardized on LOS group, what implications does it have for the interpretation of the correlation between HSMR and LOS?
--	--

3. Given that HSMR and readmission are standardized on admission category (urgent / elective), what implications does it have for the interpretation of the correlation of the number of emergency department visits with HSMR and readmission?

4. Although CIHI controlled for several potential confounding factors to estimate HSMR and readmission rates, “risk-adjustment modelling cannot entirely eliminate differences in patient characteristics among hospitals, because not all risk factors are adjusted for; if left unadjusted for (due to reasons such as viability), hospitals with the sickest patients or that treat rare or highly specialized groups of patients could still score poorly.”, as it is specified in Page 15 of the General Methodology Notes published by CIHI (Reference 14 in the manuscript). This limitation should be highlighted in the Discussion section. For example, CIHI did not include sociodemographics, primary care access, or hospital characteristics in their models.

Robustness of the methodology / Statistics / Rigor in the interpretation of some results

5. Does the CIHI provide the point estimates of risk-adjusted indicators with their 95% Confidence Intervals (CI)? The uncertainty surrounding these estimates is not taken into account in the analysis, as authors computed differences, means, paired t-tests, and correlations based on point estimates. What are the implications on the results?

6. The use of sophisticated statistical methods is limited by the aggregated data. But correlation does not imply causation, and it should be pointed out in the limitations, especially when correlation is calculated between two unadjusted variables.

7. I recommend authors to provide the 95% CI around the correlation coefficients. A coefficient of 0.40 with a 95% CI ranging from 0.10 to 0.70 would not allow a definitive conclusion about the strength of the relationship between the two variables. It is also important to check scatterplot.

8. In the Results section, please do not interpret any results without reporting outcomes values with their 95% CI and/or P-value. For example, in Page 8 Lines 4-5 (“Hospital deaths (HSMR) largely declined, while readmissions increased”). If this statement is linked to the following results in Page 8 Lines 15-18 then you should put the two sentences one after the other: “A paired-t test showed statistically significant changes in trend over time for both indicators: hospital deaths (HSMR) improved by a mean of -5.1 (95% CI, -7.33 – -2.9, t=-4.58, df=80, p<.001). And readmission worsened by a mean of 0.15% (95% CI, 0.04 – 0.26, t=2.81, df=80, p=.006).” It is not necessary to report t and df here.

9. Interpretation of the following results Page 8 Lines 6-15 is not correct, because 95% CI are overlapping. The differences observed are trends, but with not statistically significant results you cannot conclude. You should either not comment on non-significant results or indicate when results are non-significant.

- “For both indicators, the Community-Large hospital peer-group showed greater improvement than Teaching hospitals. Community-Large hospitals on average improved on in hospital deaths (HSMR) by -6.0% (95% CI -9.1 – -2.8) compared to Teaching hospitals at -4.1% (95% CI -7.5 – -0.8).”
- “Similarly, Community-Large hospitals, while increasing in readmission rates on average 1.6% (95% CI -0.3 – 3.4), had a more favourable rate than the average for Teaching hospitals at 2.1% (95% CI 0.7 – 3.6).”

• “Furthermore, for the 2017-18 data year, Community-Large hospitals had lower average rates across all three indicators of LOS, hospital deaths (HSMR), and readmission “. Here CI are overlapping except for readmission.

Objectives and added value of the study

This study has four research questions (Page 5 Lines 46-54), but I find that either the added value is not obvious, or the results are not discussed enough.

10. Objective 1 (“What are the performance trends in hospital deaths (HSMR) and readmission over time?”): Could the authors clarify what the present study adds in comparison with their previous paper [Fekri BMJ Open 2020]? Indeed, as they say in the Discussion Page 14 Lines 4-9, “Our earlier research¹⁹ established that, over time, Canadian hospitals have largely improved on in-hospital mortality; readmission rates have been trending upward; and that good or bad performance in one domain of care does not automatically reflect the same performance in other domains.”, which is the same conclusion they made in the present study. The previous work was based on the same data (except that they used 2012-17 data instead of 2013-18), at the national level and by stratifying by peer-groups of hospitals too. I suggest that the authors reformulate their objective by further emphasizing that the objective is to study how each individual hospital performed in both indicators (and not to study HSMR and readmission trends separately).

11. Objective 2: Could the authors give more arguments about the contribution of their study to the topic? Indeed there is a broad literature about the effect of LOS on risk of readmission or mortality, with some studies using more sophisticated methods (modelling and not correlations, with adjustment on several potential confounding factors) or more relevant outcomes (like potentially avoidable readmission in order to discard planned readmissions and also urgent readmissions unrelated to the index stay), with more homogeneous population (as it is mentioned in Page 4 lines 4-6: “LOS is an aggregate of all hospitalisations, and could not be restricted to condition-specific cases (of hospital death or readmission)”.

12. Objective 3: The authors enumerated the strength and the directionality of the correlation coefficients of the eight hospital characteristics with the three performance indicators in the Results section, and summarized their findings in the following way in the discussion: “Eight hospital characteristics showed trends in strength and directionality of correlation with hospital deaths (HSMR), readmission and LOS.” But what is the point? What is the underlying hypothesis? Could the authors give a meaning to these results?

13. Objective 4: Differences observed between teaching and Community-Large hospitals are not really discussed.

Minor comments

14. Abstract: I suggest renaming the “Setting” section into “Setting and Participants” and to delete the following element: “Participants: Analysis focused on indicator results and characteristics of individual Canadian hospitals.”

15. Introduction: Page 5 Line 40: “this study will use”: please use the past tense.

16. Methods

	 • Please move the following element Page 6 Lines 55-58 into the Results section: “Therefore, a total of 119 hospitals were included in the overall study, and a subset of 81 hospitals were included in the performance trend analysis.” • Please give the percentage of teaching and Community-Large Canadian hospitals that were excluded for the analysis. • Page 7 Lines 3-4 (“Descriptive statistics for the analysis of LOS, Hospital Deaths (HSMR) and Readmission indicators are presented by range of values, peer-group means and 95% confidence intervals (CI), and coefficient of variation (CoV)”: Are the variables normally distributed? If not, medians (IQR) would be more relevant. • Page 7 Lines 7-8: “A paired-t test was used to determine whether absolute changes in rates between 2013-14 and 2017-18 were significant.”: Are these differences between 2013-14 and 2017-18 normally distributed? If not, a Wilcoxon signed-rank test would be more appropriate. • Page 6 Line 39: “A hospital is designated as ‘Teaching’ by provincial/territorial ministries of health, or were identified”: Please replace “were” by “was”. 17. Results: Page 8 Lines 25-26: “While coefficient of variation values are largely similar between the two peer groups”: Please report values with their 95% CI to demonstrate this statement. 18. Tables / Figures  • Table 1: Please replace the “—” by “to” (for better readability and not to be confused with minus sign), and add a Total column and P values. • Figure 1: For a better understanding I suggest adding beside each axis the meaning (improvement / degradation) and the direction with arrows. • Table 3 and 4: Please add period of the analysis in the title (2017-2018) • Figures 2 et 3: I find it is not easy to interpret the three outcomes on the same graph, furthermore these figures are not commented on in the text. If the authors have nothing to comment on these figures, I suggest they remove them. • Table 4: Please define RIW in the footnote.
--	---

VERSION 1 – AUTHOR RESPONSE

Reviewer: 1

Reviewer Name: SANDRO SCARPELINI

Institution and Country: UNIVERSITY OF SÃO PAULO, Brazil

Please leave your comments for the authors below

1. Very good and important work. In particular, for the suggestion of a "new paradigm" for the future of the high complexity hospital's organization. As the author says that this new model has been developed since 1990, we can not forget that the study was done in Canada. In low in middle income countries, we are so far for starting this type o new organization, which transform this paper in a very interesting reference for these countries.

- We thank the reviewer for the supportive comments.

Reviewer: 2

Reviewer Name: Center for Health Technology and Services Research (CINTESIS) and Department of Community Medicine, Information and Health Decision Sciences (MEDCIDS), Faculty of Medicine, University of Porto, Porto, Portugal

Please leave your comments for the authors below

The paper addresses a relevant aspect of hospital performance assessment regarding intertwined relationship of different performance indicators and the importance of acknowledging it by policy-makers. The presentation of results is very appealing and well thought.

Major comments:

While the paper presents a strong motivation, well-referenced, to use the indicators HSMR, readmission, and LOS, assessing their association as performance measures, the focus on other hospital characteristics, in my opinion, lacks a proper motivation. Particularly, why they have compared the two types of hospitals (community vs. Teaching hospitals) is not explored in the introduction nor is discussed in the section "Reflection's on the study findings". After reading this section it seems more relevant to compare hospital merges vs. hospital not merged and rural/remote area vs. urban hospitals.

- Thank you for the helpful comment. We have added in the second paragraph of the Discussion section an explanation of why we decided to subdivide the analyses by the two hospital peer groups, and for the inclusion/investigation of hospital characteristics.
- Hospital characteristics were included in the analysis because: 1) the data were readily available for each hospital; 2) few studies had included hospital characteristics in their analyses (both as many, and as diverse as the ones available from this dataset); and 3) in doing so, it offered an opportunity to explore and observe any associations between the outcome indicators (subdivided by hospital peer groups).
- With respect to hospital mergers, there are two reasons why we cannot explore this type analyses: 1) in the event of a hospital merger, there is a natural disruption in the ability to perform trending analyses; 2) hospital mergers do not occur frequently enough to have a sufficient sample size for analyses. Lastly, the designation of whether a facility is located in a rural/remote or urban geographic location was not available in the dataset.

In the trend analysis of HSMR and readmissions, why did you use just the difference between the first and last period available and not the longitudinal analysis using several measures per hospital over the 5 years? Would that give you different results?

- Thank you for the suggestion. We chose a paired t-test to determine the direction and potential significance of any trend when comparing 2013-14 and 2017-18 data years. The results showed a significant decrease in rates for hospital deaths (HSMR), and a significant increase in readmission rates (paragraph 1, Results section).
- Along the lines of your suggestion, we performed a linear regression analysis with indicator results as the outcome and year as the predictor for all hospital values for the 5 years, and our results were in line with the paired-t test results.

One of the strengths of the study is the quality and comprehensiveness of the data used, which includes several indicators and covers most of the Canadian hospitals. However, in my opinion, the data has also important limitations that have not been addressed in the discussion. Specifically,

- The indicator All Patients Readmitted to Hospital does not control for the hospital case mix, and therefore, this could explain for example why teaching hospitals have a larger increase regarding readmissions; or could affect the association between indicators.
- Thank you for the important comment. We have elaborated in the Methods section the concept of peer groups, and the importance of only comparing hospital performance within each peer-group. While hospital structural characteristics are not included for in risk-adjustment models, patient case

mix is included. Therefore, as you rightfully pointed out, comparing the individual indicator values of a Teaching hospital and Community-Large hospital would not be valid.

- I am not familiar with the readmissions indicator, but I believe that if a patient dies during the 30-day period after discharge, the hospitalization will be counted in the denominator of this indicator. However, since this is a competing risk it cannot be counted in the numerator since the patient can no longer be readmitted. This may attenuate the increase of readmissions.
- In-hospital deaths may not be the best performance indicator. It is certainly important for the period the patient is in the hospital, but 30-day mortality is considered a more suited indicator since the death during this period is likely to be associated with the care provided at the hospital.
- Thank you for these two related comments. We have added a paragraph in the Strengths and Limitations section describing the limitations of both indicators due to the inability to include data from deaths that occur outside of hospital (see paragraph prior to Reflections on the study's findings).

There seems to be an outlier community hospital in Figure 3, long stays, high death rate, and average readmission rate. Do you know why? Could you please elaborate or make a remark on this case in the discussion. Does it influence the correlation estimate?

- Thank you for the important observation. We have added a note in the Discussion section on our continued inclusion of this outlier hospital in Figure 3. As per a comment by Reviewer 3, we are omitting the correlation analysis of Hospital Deaths (HSMR) and LOS (as it is a risk-adjustment variable in the model). Therefore, we do not believe it will strongly influence the correlation estimate between an average readmission rate and a high hospital death (HSMR) rate.

Minor comments:

Page 8, Line 15: "A paired-t test showed statistically significant changes in trend over time for both indicators: hospital deaths (HSMR) improved by a mean of -5.1 (95% CI, -7.33 – -2.9, $t=-4.58$, $df=80$, $p<.001$). And readmission worsened by a mean of 0.15% (95% CI, 0.04 – 0.26, $t=2.81$, $df=80$, $p=.006$)." Does this refer to all hospitals? If yes, I think it would be better in line 2 of this page.

- Thank you for this observation. We have now placed the sentences earlier in the section.

Table 2 would read better if both the first and second columns were sorted by "green arrow", "red arrow" and "equal sign". As is, only the first column is sorted.

- Thank you for the suggestion. Both columns are now sorted accordingly.

Even though it is stated in the methods, I recommend adding the reference period of the analysis (2017-18) in the subtitle of the results in Page 10, line 36, and on Page 12, line 1.

- Thank you for the suggestion. The reference year has been added to both locations.

For comparisons between Figures 2 and 3 the shade of the markers (based on the LOS) should have the same scale. It seems that an average LOS of 7.7 in figure 2 is darker than in Figure 3.

- Thank you for the observation. The LOS legend gradient is now constant across the two Figures.

Please provide the abbreviation for RIW in table 4, page 12. Resource Intensity Weight?

- Thank you for the observation. The acronym for RIW has been corrected.

I prefer to see the strengths and limitations at the bottom of the discussion, but it may be just a personal preference.

- Thank you for the suggestion. Our previous publication with BMJopen followed our existing format. We feel it provides for a more informed reflection on the study's findings having just discussed the strengths and limitations of the study just prior.

Reviewer: 3

Reviewer Name: Léa Pascal
Institution and Country: Hospices Civils de Lyon, France
Please leave your comments for the authors below

Overall comment

Using aggregated hospital-level data from 119 teaching and large-sized Canadian hospitals, Fekri O. and colleagues evaluated the evolution of hospital death and readmission between 2013-14 and 2017-18. They also studied the correlations of eight hospital characteristics with hospital death, readmission, and length of stay (LOS) in 2017-18. They concluded that care must be taken in the use of performance indicators to establish public health policies, as a hospital may have good performance with one indicator but poor performance with another. The subject is interesting, and a strength of this study is the use of publicly reported hospital performance indicators at the national level.

However, important details concerning the methodology used for producing the hospital performance indicators are lacking while they are necessary to the interpretation of results. I also have concerns about the robustness of the methodology and the rigor in the interpretation of some results. Lastly, I would like the authors clarify what this study adds to the literature and in particular to a previous study published on the same data concerning one of their four research questions [Fekri O, Manukyan E, Klazinga N. Appropriateness, effectiveness, and safety of care delivered in Canadian hospitals: a longitudinal assessment on the utility of publicly reported performance trend data between 2012-2013 and 2016-2017. *BMJ Open* 2020; O:e035447. Doi:10.1136/bmjopen-2019-035447].

Please find below the detailed comments, hoping that my suggestions will help the authors to improve their paper.

- We thank the reviewer for the very helpful and detailed comments.

Major comments

Methodological details behind the indicators and implications for the interpretation

Outcomes used in this study are not precisely defined in the manuscript, and the reader is invited Page 6 Line 31 to consult the technical notes of the Canadian Institute for Health Information (CIHI) (references 14 and 15) from which the performance indicators originated. It is important for the reader to have a straightforward information about the inclusion and exclusion criteria and the models and covariates used, and to be able to take these elements into account when interpreting the results and making comparisons.

1. I recommend the authors to precise in the Methods section that hospital death and readmission are risk-adjusted, and to give the covariates used in the calculation, i.e.:

- for hospital deaths (HSMR): diagnosis groups, age, sex, LOS group, admission category (urgent and elective), comorbidity (Charlson Index Score) group and transfers (with reference to Canadian Institute for Health Information. Model Specifications — Clinical Indicators, May 2020. Ottawa, ON: CIHI; 2020),
- for readmission: age, sex, acute care hospitalization in previous 6 months, admission category, comorbidity, and a selection of case-mix groups (CMG) according to patient groups considered (obstetric, pediatric, surgical, medical), with reference to Canadian Institute for Health Information. Hospital Standardized Mortality Ratio - Technical Notes, September 2019. Ottawa, ON: CIHI; 2019.
- Thank you for this recommendation. We have now described in the Methods section (Definition of Variables) the risk-adjustment variables for both indicators. We also cite the two relevant 2019 model

specifications documents by the data steward which details precise calculation methods and coefficients used in models (see paragraph 4).

2. Given that HSMR is standardized on LOS group, what implications does it have for the interpretation of the correlation between HSMR and LOS?

- Thank you for the very important observation. Since HSMR is standardized on LOS, we have now excluded it from correlation analyses (namely Table 3).

3. Given that HSMR and readmission are standardized on admission category (urgent / elective), what implications does it have for the interpretation of the correlation of the number of emergency department visits with HSMR and readmission?

- Thank you for the observation. The proportion of patients admitted through the emergency department (ED) with an admission category of Urgent or Emergent is <9% of all ED visits (source: CIHI QuickStats, ED visits: volumes and median LOS, by triage level, visit, 2017-18). Therefore, this small proportion should not have had a major impact on the interpretation of the correlation results (if it did, one would have expected the relationship should have been significant, which they are not).

4. Although CIHI controlled for several potential confounding factors to estimate HSMR and readmission rates, “risk-adjustment modelling cannot entirely eliminate differences in patient characteristics among hospitals, because not all risk factors are adjusted for; if left unadjusted for (due to reasons such as viability), hospitals with the sickest patients or that treat rare or highly specialized groups of patients could still score poorly.”, as it is specified in Page 15 of the General Methodology Notes published by CIHI (Reference 14 in the manuscript). This limitation should be highlighted in the Discussion section. For example, CIHI did not include sociodemographics, primary care access, or hospital characteristics in their models.

- Thank you for this observation. We have now added in the Strengths and Limitations section (see last paragraph) a disclaimer on the limitations of risk-adjustment.

Robustness of the methodology / Statistics / Rigor in the interpretation of some results

5. Does the CIHI provide the point estimates of risk-adjusted indicators with their 95% Confidence Intervals (CI)? The uncertainty surrounding these estimates is not taken into account in the analysis, as authors computed differences, means, paired t-tests, and correlations based on point estimates. What are the implications on the results?

- Thank you for this comment. 95% CI are available for the hospital deaths (HSMR) and readmission indicators, however, we caution their inclusion in the tests.
- These are in general large hospitals with stable patient volumes. We do not expect any significant changes in hospital characteristics over time (i.e., same peer-group designation, hospital capacity and structural make-up). In turn, we expect these estimates to be stable.
- If we were to incorporate standard deviation and calculate a z-score, we would essentially be moving to a totally different (imaginary) scale, and lose the ability to interpret the indicator results in the context of the results themselves. We would therefore be unable to explain, in a meaningful way, the extent to which hospitals may have improved or declined on these indicators.

6. The use of sophisticated statistical methods is limited by the aggregated data. But correlation does not imply causation, and it should be pointed out in the limitations, especially when correlation is calculated between two unadjusted variables.

- Thank you for this comment. We have added this caution in the limitations of the study.

7. I recommend authors to provide the 95% CI around the correlation coefficients. A coefficient of 0.40 with a 95% CI ranging from 0.10 to 0.70 would not allow a definitive conclusion about the strength of the relationship between the two variables. It is also important to check scatterplot.

- Thank you for this recommendation. We have now added 95% CI for correlation coefficients in Tables 3 and 4.

8. In the Results section, please do not interpret any results without reporting outcomes values with their 95% CI and/or P-value. For example, in Page 8 Lines 4-5 (“Hospital deaths (HSMR) largely declined, while readmissions increased”). If this statement is linked to the following results in Page 8 Lines 15-18 then you should put the two sentences one after the other: “A paired-t test showed statistically significant changes in trend over time for both indicators: hospital deaths (HSMR) improved by a mean of -5.1 (95% CI, -7.33 – -2.9, t=-4.58, df=80, p<.001). And readmission worsened by a mean of 0.15% (95% CI, 0.04 – 0.26, t=2.81, df=80, p=.006).” It is not necessary to report t and df here.

- Thank you for these comments; we have incorporated them accordingly to the first paragraph of the Results section.

9. Interpretation of the following results Page 8 Lines 6-15 is not correct, because 95% CI are overlapping. The differences observed are trends, but with not statistically significant results you cannot conclude. You should either not comment on non-significant results or indicate when results are non-significant.

- For both indicators, the Community-Large hospital peer-group showed greater improvement than Teaching hospitals. Community-Large hospitals on average improved on in hospital deaths (HSMR) by -6.0% (95% CI -9.1 – -2.8) compared to Teaching hospitals at -4.1% (95% CI -7.5 – -0.8).”

- “Similarly, Community-Large hospitals, while increasing in readmission rates on average 1.6% (95% CI -0.3 – 3.4), had a more favourable rate than the average for Teaching hospitals at 2.1% (95% CI 0.7 – 3.6).”

- “Furthermore, for the 2017-18 data year, Community-Large hospitals had lower average rates across all three indicators of LOS, hospital deaths (HSMR), and readmission “.

- Thank you for these recommendations. The paragraph containing these results have been adjusted per your comments.

Objectives and added value of the study

This study has four research questions (Page 5 Lines 46-54), but I find that either the added value is not obvious, or the results are not discussed enough.

10. Objective 1 (“What are the performance trends in hospital deaths (HSMR) and readmission over time?”): Could the authors clarify what the present study adds in comparison with their previous paper [Fekri BMJ Open 2020]? Indeed, as they say in the Discussion Page 14 Lines 4-9, “Our earlier research¹⁹ established that, over time, Canadian hospitals have largely improved on in-hospital mortality; readmission rates have been trending upward; and that good or bad performance in one domain of care does not automatically reflect the same performance in other domains.”, which is the same conclusion they made in the present study. The previous work was based on the same data (except that they used 2012-17 data instead of 2013-18), at the national level and by stratifying by peer-groups of hospitals too. I suggest that the authors reformulate their objective by further emphasizing that the objective is to study how each individual hospital performed in both indicators (and not to study HSMR and readmission trends separately).

- Thank you for suggestion. We have now revised our first research question to read “how have hospitals performed in both the hospital deaths (HSMR) and readmission indicators over time?”.

- Our previous paper largely assessed the utility of categorical hospital performance trend data (‘improving, worsening, or no change over time’), whereas our present study aims to explore the quantitative association across three interrelated indicators and facility characteristics. Indeed, the same conclusion can be drawn on broad national trends of how hospitals have performed on hospital deaths (HSMR) and readmission; however, this is only a small component on the present study. Furthermore, our examination of hospital characteristics and LOS is new in this paper, as well as the

comprehensive 'All Patients Readmitted to Hospital' (as opposed to our previous study that examined the four readmission breakdowns individually).

11. Objective 2: Could the authors give more arguments about the contribution of their study to the topic? Indeed there is a broad literature about the effect of LOS on risk of readmission or mortality, with some studies using more sophisticated methods (modelling and not correlations, with adjustment on several potential confounding factors) or more relevant outcomes (like potentially avoidable readmission in order to discard planned readmissions and also urgent readmissions unrelated to the index stay), with more homogeneous population (as it is mentioned in Page 4 lines 4-6: "LOS is an aggregate of all hospitalisations, and could not be restricted to condition-specific cases (of hospital death or readmission)").

- Thank you for this comment. We have added a paragraph (second last) in the Strengths and Limitations section on the analytical limitations of our study using aggregate hospital-level data. Furthermore, we've added this limitation to the abstract "strengths and weaknesses of this study" box. Paragraph 2 of the Discussion section, with a similar comment from Reviewer 2, now further elaborates on our study.

12. Objective 3: The authors enumerated the strength and the directionality of the correlation coefficients of the eight hospital characteristics with the three performance indicators in the Results section, and summarized their findings in the following way in the discussion: "Eight hospital characteristics showed trends in strength and directionality of correlation with hospital deaths (HSMR), readmission and LOS." But what is the point? What is the underlying hypothesis? Could the authors give a meaning to these results?

- Thank you for this comment. We have noted now in the Discussion section (last paragraph prior to Strengths and Limitations section) that since our study was exploratory in nature, using the type of data at hand and incorporating hospital characteristics, we did not begin with an explicit hypothesis on any potential association.

13. Objective 4: Differences observed between teaching and Community-Large hospitals are not really discussed.

- Thank you for this observation. In elaborating the results for objectives 1, 2 and 3, we have parsed all results by Teaching and Community-Large hospitals. We chose this approach rather than having only 3 research questions but listing "... and the differences between Teaching and Community-Large hospitals" at the end of each research question. We do however appreciate this comment, and have elaborated on differences between the two peer groups in the second paragraph of the Discussion.

Minor comments

14. Abstract: I suggest renaming the "Setting" section into "Setting and Participants" and to delete the following element: "Participants: Analysis focused on indicator results and characteristics of individual Canadian hospitals."

- Thank you for the suggestion. However, these two distinct section headings are per BMJopen abstract requirements. (<https://bmjopen.bmj.com/pages/authors/>)

15. Introduction: Page 5 Line 40: "this study will use": please use the past tense.

- Thank you for the suggestion. We have now corrected the sentence to the past tense.

16. Methods

- Please move the following element Page 6 Lines 55-58 into the Results section: "Therefore, a total of 119 hospitals were included in the overall study, and a subset of 81 hospitals were included in the performance trend analysis."

- Thank you for the suggestion. We believe this sentence describes the sample of the study, and not a result of the analysis. This was a pre-determined criterion for the analysis. We would prefer to leave this information in the Methods section.

- Please give the percentage of teaching and Community-Large Canadian hospitals that were excluded for the analysis.

- Thank you for this suggestion. We have now added the percentage of hospitals included in the analysis for each peer-group (paragraph 7 of Methods section).

- Page 6 Line 39: "A hospital is designated as 'Teaching' by provincial/territorial ministries of health, or were identified": Please replace "were" by "was".

- Thank you for this suggestion. The edit is now reflected.

- Page 7 Lines 3-4 ("Descriptive statistics for the analysis of LOS, Hospital Deaths (HSMR) and Readmission indicators are presented by range of values, peer-group means and 95% confidence intervals (CI), and coefficient of variation (CoV)": Are the variables normally distributed? If not, medians (IQR) would be more relevant.

- Thank you for this comment and suggestion. The variables are normally distributed, and we've added IQR and medians to the table.

- Page 7 Lines 7-8: "A paired-t test was used to determine whether absolute changes in rates between 2013-14 and 2017-18 were significant.": Are these differences between 2013-14 and 2017-18 normally distributed? If not, a Wilcoxon signed-rank test would be more appropriate.

- Thank you for this comment. A strength of the paired t test in this context is that each hospital becomes its own control, thereby making the distribution of the differences more symmetrical. And thus, the central limit theorem should apply here. Histograms showing the differences of the two years for each indicator are symmetrical.

17. Results: Page 8 Lines 25-26: "While coefficient of variation values are largely similar between the two peer groups": Please report values with their 95% CI to demonstrate this statement.

- Thank you for this comment. We have now added 95% CI to CoV values in Table 1.

18. Tables / Figures

- Table 1: Please replace the "—" by "to" (for better readability and not to be confused with minus sign), and add a Total column and P values.

- Thank you for these suggestions. We have replaced "—" with "to" in Table 1. We do not feel a 'Total' column is necessary as the only cumulative data would be the two hospital peer-groups (36+45=81) which is described in the Methods section. Lastly, there are no p values to insert into Table 1. Relevant tests with p values are described in the text.

- Figure 1: For a better understanding I suggest adding beside each axis the meaning (improvement / degradation) and the direction with arrows.

- Thank you for this suggestion. Directional arrows depicting improving/worsening performance is now added to Figure 1.

- Table 3 and 4: Please add period of the analysis in the title (2017-2018)

- Thank you for this suggestion. The period of analysis has now been added to the titles of both tables.

- Figures 2 et 3: I find it is not easy to interpret the three outcomes on the same graph, furthermore these figures are not commented on in the text. If the authors have nothing to comment on these figures, I suggest they remove them.

- Thank you for the comment. We have now aligned the colour scale legends for the two figures (as per the comment from Reviewer 2) to improve the ability to interpret and compare the two peer-groups.
- Table 4: Please define RIW in the footnote.
- Thank you for this observation. We have now placed a footnote in the table for the RIW acronym.

Editorial requests:

- Please note that the reviewers have raised a number of major concerns with your study. As there are substantial concerns with your manuscript, we will be seeking further advice when we receive your revisions. We cannot, at this stage, anticipate the final decision on your manuscript.
- Along with your revised manuscript, please include a copy of the RECORD checklist indicating the page/line numbers of your manuscript where the relevant information can be found (<http://www.record-statement.org/>).

VERSION 2 – REVIEW

REVIEWER	Mariana Lobo Faculty of Medicine, University of Porto
REVIEW RETURNED	16-Nov-2020

GENERAL COMMENTS	In the revised version of the manuscript, the authors have correctly addressed the comments and suggestions. Thank you
--

REVIEWER	Léa Pascal Hospices Civils de Lyon, France
REVIEW RETURNED	22-Nov-2020

GENERAL COMMENTS	I thank the authors for their work in revising the manuscript. Comments and suggestions have been addressed. I think this revised version has improved the manuscript. I have only a few very minor suggestions (see list below, with the page and line numbers based on the revised proof).  1. In the Results section, Page 8 Lines 4-9, as suggested the authors have rearranged the order of the sentences, which makes this paragraph clearer in my opinion. However they also changed the initial terms “largely declined” to “showed a mean improvement of lowering rates”, and “increased” to “showed a mean increase in rates”, which seems less understandable to me. I suggest keeping the initial terms (“largely declined” and “increased”). 2. Page 10 Line 42: Please report 95% CI around the CoV. 3. Page 10 Lines 41-46: “While Community-Large hospitals showed [...] compared to Teaching hospitals [...], their mean peer group LOS values were still lower than Teaching hospitals (6.5 days compared to Teaching hospitals at 7.1) (see table 1). Mean LOS of patients in Community-Large hospitals was 0.6 days shorter, or roughly half a day, compared to Teaching hospitals (6.5 vs. 7.1 days)”: Please could you remove the redundancy between these two sentences? Could you also report 95% CI around these mean LOS values, and specify that the difference between Community-Large hospitals and Teaching hospitals is not statistically significant as 95% CI are overlapping?
--

	4. Thank you for adding medians (IQR) in Table 1. I find, however, that it would be more informative to present (1st Quartile – 3rd Quartile) rather than the difference between them. 5. Please could you report 95% CI around the correlation coefficients in the part of the text that describes the results of Table 4 (Page 14 Lines 13-38)? 6. In the Methods/Definition of variables section, Page 6 Line 40, I am not sure, but is the word "for" correct here? (“[...] this range of risk-adjustment variables for are: age, sex [...]”)
--	--

VERSION 2 – AUTHOR RESPONSE

Reviewer: 2

Reviewer Name: Mariana Lobo

Institution and Country: Faculty of Medicine, University of Porto

Please state any competing interests or state 'None declared': None declared

Comments to the Author

In the revised version of the manuscript, the authors have correctly addressed the comments and suggestions. Thank you.

- We thank the reviewer for the comments.

Reviewer: 3

Reviewer Name: Léa Pascal

Institution and Country: Hospices Civils de Lyon, France

Please state any competing interests or state 'None declared': None declared

Comments to the Author

I thank the authors for their work in revising the manuscript. Comments and suggestions have been addressed. I think this revised version has improved the manuscript.

- We thank the reviewer for the time and attention in providing these additional minor suggestions for improvement.

I have only a few very minor suggestions (see list below, with the page and line numbers based on the revised proof).

1. In the Results section, Page 8 Lines 4-9, as suggested the authors have rearranged the order of the sentences, which makes this paragraph clearer in my opinion. However they also changed the initial terms “largely declined” to “showed a mean improvement of lowering rates”, and “increased” to “showed a mean increase in rates”, which seems less understandable to me. I suggest keeping the initial terms (“largely declined” and “increased”).

- Thank you for this suggestion – we have returned the initial terms.

2. Page 10 Line 42: Please report 95% CI around the CoV.

- Thank you for this suggestion – 95% CI are now noted in this sentence.

3. Page 10 Lines 41-46: “While Community-Large hospitals showed [...] compared to Teaching hospitals [...], their mean peer group LOS values were still lower than Teaching hospitals (6.5 days compared to Teaching hospitals at 7.1) (see table 1). Mean LOS of patients in Community-Large hospitals was 0.6 days shorter, or roughly half a day, compared to Teaching hospitals (6.5 vs. 7.1 days)”: Please could you remove the redundancy between these two sentences? Could you also report 95% CI around these mean LOS values, and specify that the difference between Community-Large hospitals and Teaching hospitals is not statistically significant as 95% CI are overlapping?

- Thank you for these two suggestions – we have applied them accordingly in the paragraph.

4. Thank you for adding medians (IQR) in Table 1. I find, however, that it would be more informative to present (1st Quartile – 3rd Quartile) rather than the difference between them.

- Thank you for this suggestion – we now present Q1 and Q3 quartiles accordingly in Table 1.

5. Please could you report 95% CI around the correlation coefficients in the part of the text that describes the results of Table 4 (Page 14 Lines 13-38)?

• Thank you for this suggestion – we now display 95% confidence intervals when discussing these results.

6. In the Methods/Definition of variables section, Page 6 Line 40, I am not sure, but is the word "for" correct here? (“[...] this range of risk-adjustment variables for are: age, sex [...]”)

• Thank you for this suggestion – the word ‘for’ has now been removed.

VERSION 3 – REVIEW

REVIEWER	Léa Pascal Hospices Civils de Lyon, France
REVIEW RETURNED	17-Dec-2020
GENERAL COMMENTS	I thank the authors for taking these last minor comments into account. I have no further comments.